# Effect of Heat Treatment on Microstructure and Mechanical Properties of Titanium Alloy Fabricated by Laser–Arc Hybrid Additive Manufacturing

**Yuhang Chen [1], Juan Fu [1,*], Lilong Zhou [2], Yong Zhao [1], Feiyun Wang [1], Guoqiang Chen [3] and Yonghui Qin [3]**

[1] Provincial Key Laboratory of Advanced Welding Technology, Jiangsu University of Science and Technology, Zhenjiang 212003, China; nick_cyh@163.com (Y.C.); yongzhao418@just.edu.cn (Y.Z.); wangfeiyunmse@126.com (F.W.)

[2] Aerospace Engineering Equipment (Suzhou) Co., Ltd., No. 81 North Guandu Road, Wuzhong District, Suzhou 215104, China; zhoulilongchn@foxmail.com

[3] Jiangsu Yangzi-Mitsui Shipbuilding Co., Ltd., Taicang 215400, China; chenguoqiang@yzjship.com (G.C.); qyh@yzjship.com (Y.Q.)

\* Correspondence: fujuan@just.edu.cn

**Abstract:** The tailored thermal heat-treatment process for Ti-6Al-4V alloy manufactured by laser–arc hybrid additive manufacturing can achieve desired microstructures and excellent mechanical properties for components. The effects of different heat treatment regimens on the microstructure and mechanical properties of Ti-6Al-4V alloy manufactured by laser–arc hybrid additive manufacturing are investigated in this study. Utilizing optical microscopy and scanning electron microscopy, we analyze the variations in microstructure with changes in heat-treatment parameters and explore the reasons for the changes in mechanical properties under different solutions' treatment temperatures and cooling rates. The microstructure of Ti-6Al-4V alloy fabricated via laser–arc hybrid additive manufacturing was primarily composed of Widmanstätten α plate structures and a small amount of acicular martensite α′ within columnar β grains that grew outward from the substrate along the deposition direction. Following solution treatment and aging heat treatment, the microstructure transitioned to a typical high-performance net basket structure with significantly reduced α plate thickness, leading to noticeable enhancements in sample ductility and toughness. Specifically, when the solution treatment and aging treatment regimen was set at 950 °C for 1 h, followed by air cooling, and then aging at 540 °C for 6 h with subsequent air cooling, the average grain size decreased by a factor of two compared to the as-deposited samples, while the impact toughness increased by 66.7%.

**Keywords:** laser–arc hybrid additive manufacturing; Ti-6Al-4V alloy; heat treatment; microstructure; mechanical properties

## 1. Introduction

Ti-6Al-4V alloy, which is known for its excellent performance as a dual-phase titanium alloy, has found widespread applications in a diverse range of fields, including marine vessels, aerospace, and biological research [1]. With the rapid development of manufacturing industries, traditional forming processes struggle to meet the demand for rapid one-time shaping of large structural components [2]. Wire arc additive manufacturing (WAAM) processes provide significant advantages for large component production due to their efficiency, cost-effectiveness, and flexibility [3]. However, WAAM processes suffer from issues such as low forming accuracy and poor material utilization. Laser–arc hybrid additive manufacturing (LAHAM) has garnered considerable attention due to its ability to improve both forming accuracy and production efficiency [4]. Compared to WAAM, LAHAM processes exhibit smaller melt pool sizes and lower temperature gradients at the same scanning speeds, resulting in higher forming accuracy and substantial potential applications [5]

The microstructure of components fabricated using LAHAM is relatively fine, leading to enhanced mechanical properties [6]. In the welding domain, the coupling effect of laser and arc has been shown to stabilize arc behavior while improving the microstructural properties of weld seams. Scholars have begun applying this technique in additive manufacturing. Liu et al. reported that compared to WAAM, LAHAM reduces Zn element vaporization by 5.8%, refines grain size by approximately twofold, and achieves a more uniform distribution of related elements. These improvements resulted in an 11.4% increase in ultimate tensile strength and a 29.9% increase in yield strength [7]. Gao et al. demonstrated that, under the same maximum wall thickness conditions, LAHAM reduces surface roughness by 34.7% compared to WAAM samples, increases effective wall thickness by 20%, and exhibits lower anisotropy [8]. Gong et al. observed that increasing the laser power enhanced the stability of the melt pool during the additive process, but this stability subsequently decreased. Similarly, the surface accuracy of the melt pool initially improved but later deteriorated [9].

It should be noted that the rapid solidification and multiple rapid annealing cycles inherent to the LAHAM process and the microstructure and residual stress distribution of LAHAM Ti-6Al-4V alloy differ significantly from those of forged or cast Ti-6Al-4V alloys [10]. However, there is limited research on the microstructural changes that occur in titanium alloy after additive manufacturing heat treatment [11–15], and our understanding of the microstructural evolution and mechanical properties of LAHAM Ti-6Al-4V alloy, particularly after heat treatment, remains limited, which hinders the development and application of this material. Therefore, this paper primarily investigates the influence of solution treatment parameters such as solution temperature and cooling methods, on the microstructure and mechanical properties of LAHAM Ti-6Al-4V alloy.

## 2. Materials and Experimental Methods

As illustrated in Figure 1, the LAHAM experimental system was utilized to prepare samples using a LAHAM process. The system comprises an IPG 10kW fiber laser (IPG Photonics Corporation, Newton, Massachusetts, USA), a KUKA six-axis robotic arm, and a Fronius TPS500i single-wire welding machine (Wels, Austria). The arc heat source in this experiment was mainly provided by a TPS500i welding machine manufactured by Fronius. The diameter of the Ti-6Al-4V metal wire (FuShiTe, Baoji, China) was 1.2 mm. During additive manufacturing, the arc current was 178 A, the arc voltage was 21.02 V, the wire feed speed was 6.0 m/min, and the scanning speed was 0.6 m/min. The laser beam was perpendicular to the scanning direction, the included angle between the MIG welding gun and the laser head was 45°, the spot size of the laser at a defocus of 0 was 2 mm, and the distance from the laser to the arc was 2 mm. The experiments were conducted within a controlled atmosphere chamber filled with argon gas. To ensure precision in sample formation due to differences in arc ignition and extinguishing phases, a reciprocating scanning deposition method was employed.

A 12 mm thick Ti-6Al-4V alloy plate was selected as the substrate. Prior to LAHAM processing, the substrate surface was prepared by grinding off the oxide layer using hard alloy files, followed by polishing with sandpaper and cleaning with acetone before LAHAM processing. The LAHAM process parameters were set as follows: laser power, 1500 W, wire feed rate, 6.0 m/min, scanning speed, 0.6 m/min, and argon gas flow rate, 100 L/min. The fabricated LAHAM Ti-6Al-4V alloy samples measured 350 mm × 100 mm × 7 mm.

Microscopic observations of the microstructure were conducted using a ZEISS Axio Observer optical microscope (Oberkochen, Germany), while sample microstructural characterization was performed using a Regulus-8100 scanning electron microscope (ZEISS, Oberkochen, Germany). Tensile tests were conducted at room temperature using a CMT5205 electronic universal tensile (Jinan Precision Testing Equipment Co. Ltd., Jinan, China) testing at a loading rate of 2 mm/min. Two tensile tests were performed under each test condition, and the average value of the test results was taken to ensure the accuracy of the test, while impact tests were carried out using a SANS pendulum impact tester (Jinan,

China). The impact test specimens were prepared in accordance with the standard method of the metal Charpy v-notch impact test. The dimensions of the impact test specimens are shown in the figure. At room temperature, an impact test was performed using a ZBC-2302-D pendulum impact tester (SANS, Jinan, China). The impact energy of the pendulum was 150 J. Three impact tests were performed under each test condition, and the average value of the three tests was taken as the experimental result to ensure the accuracy of the test results.

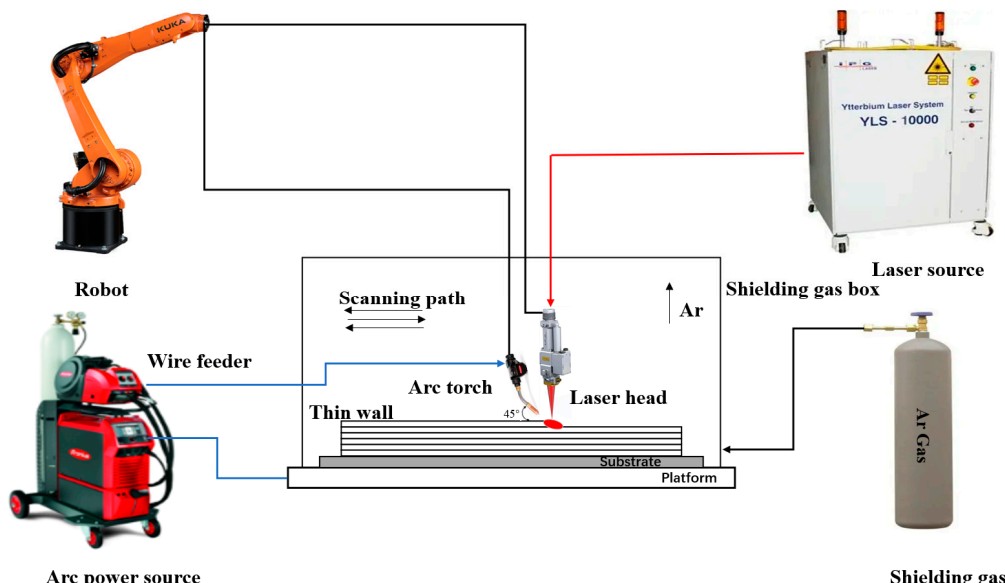

**Figure 1.** Schematic diagram of LAHAM system.

Additionally, microscopic hardness tests were conducted on thin-walled specimens that had undergone rough grinding, fine grinding, and fine polishing using a German KB fully automatic Vickers hardness tester (KB Prüftechnik Gmb, Lügde, Germany). Microscopic hardness tests were performed at room temperature in the stable region of the additive specimen's cross-section in the vertical direction of the additive. The number of indents for the microscopic hardness test was 50, the spacing between indents was 1 mm, the applied load was 5 N, and the holding time was 10 s. A schematic diagram showing the sampling positions for the tensile specimen, impact specimen, and metallographic specimen are shown in Figure 2. The geometric dimensions of the tensile and impact specimens are shown in Figure 3.

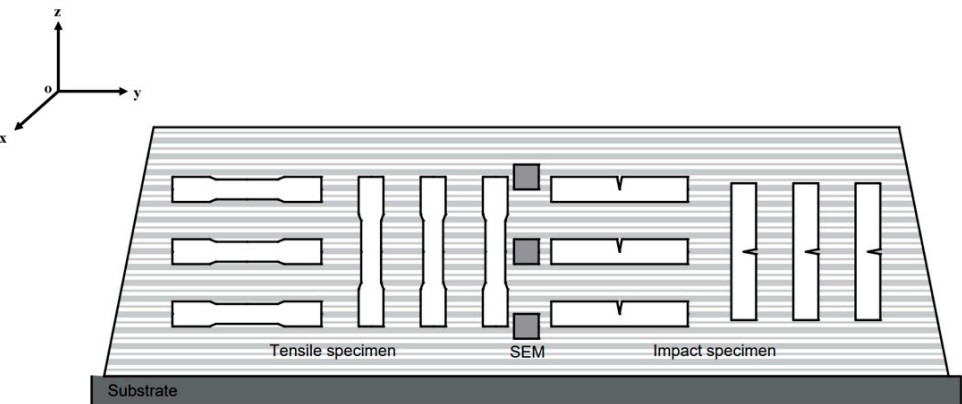

**Figure 2.** Schematic diagram of the sampling positions for the tensile specimen, impact specimen, and metallographic specimen.

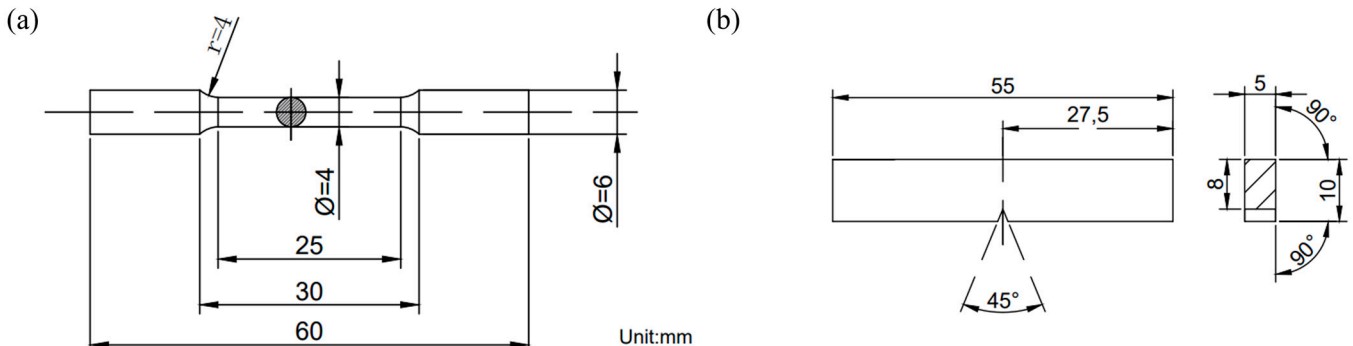

**Figure 3.** Schematic diagram: (**a**) tensile test specimen; (**b**) impact test specimen dimensions.

The heat treatment regimens and parameters employed for the LAHAM Ti-6Al-4V alloy samples are detailed in Figure 4 and Table 1. During the heat treatment of α + β titanium alloy, the size, shape, and distribution of the α phase will affect the β phase. Therefore, the solid solution temperature is generally selected to be within 100 °C below the β phase transition temperature (998 °C ± 100 °C). Based on this, for this experiment, we set the solid solution temperatures to 900 °C, 950 °C, and 1000 °C, and the solid solution time to 1 h. The solid solution process of 950 °C/1 h with relatively good observation and performance was selected to study and discuss the differences in the microstructure and performance of LAHAM Ti-6Al-4V alloy under different cooling methods after "solid solution + aging". Aging treatment can release the lattice strain energy generated after the solid solution, promote the precipitation of the second phase, and decompose the unstable phase. The selection of aging temperature is generally above 500 °C and below the β phase transition point (1000 °C), which can effectively prevent the appearance of the ω phase. As a transition phase between the α phase and β phase, the ω phase has high hardness and brittleness, which will shorten the service life of the workpiece. Based on this, this experiment selected an aging temperature of 540 °C and a holding time of 6h and finally determined the heat treatment process of solid solution + aging at 900 °C, 950 °C, and 1000 °C. To investigate the effects of solution temperatures on microstructure and mechanical properties, the samples were subjected to 1 h solution treatment at 900 °C, 950 °C, and 1000 °C, followed by air cooling. Subsequently, they were maintained at a temperature of 540 °C for 6 h prior to air cooling. To study the influence of the cooling rate on microstructure and mechanical properties, another set of samples underwent 1 h solution treatment at 950 °C followed by furnace cooling, air cooling, and water quenching, These samples were then held at 540 °C for 6 h before air cooling. This experimental design allowed for the elucidation of the interrelationships between the solution temperature, cooling rate, microstructure, and mechanical properties.

**Table 1.** Heat-treatment parameters for LAHAM of Ti-6Al-4V alloy.

| Sample | Heat Treatments * |
|--------|-------------------|
| 1 | 900 °C/1 h, AC + 540 °C/6 h, AC |
| 2 | 950 °C/1 h, AC + 540 °C/6 h, AC |
| 3 | 1000 °C/1 h, AC + 540 °C/6 h, AC |
| 4 | 950 °C/1 h, FC + 540 °C/6 h, AC |
| 5 | 950 °C/1 h, WQ + 540 °C/6 h, AC |

* WQ—Water quenching; FC—furnace cooled; AC—air cooled.

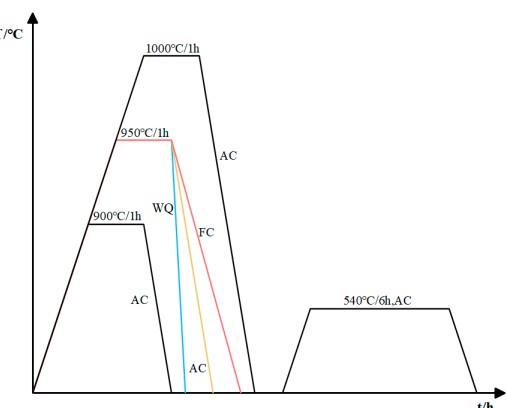

**Figure 4.** Heat-treatment regimens for solution treatment and aging treatment of LAHAM Ti-6Al-4V alloy samples.

## 3. Results and Discussion

### 3.1. As-Deposited Microstructures

Figure 5 shows the surface formation of an as-deposited thin-walled wall. Figure 6a presents the typical macroscopic morphology of LAHAM Ti-6Al-4V alloy, exhibiting continuous extrusion growth of β columnar crystals traversing multiple deposition layers [16]. The average width of the columnar crystals is approximately 350 μm, with α lamellae averaging 2.9 μm in thickness. Due to differing crystal orientations, the macroscopic structure of the deposited walls presents distinct bright and dark banding features. During the LAHAM process, significant temperature gradients develop within the melt pool, perpendicular to the scanning direction. Solidification commences at the bottom of the melt pool, and due to the narrow solidification interval characteristic of Ti-6Al-4V alloy, equiaxed crystal structures struggle to form within the deposition walls. The microstructure within β grains mainly comprises fine Widmanstätten structures, as illustrated in Figure 6b. Bright layering between deposition layers is observed, and this attributed to the reheating effect from subsequent deposition layers, resulting in thermal coarsening of α grains within the heat-affected zone [17].

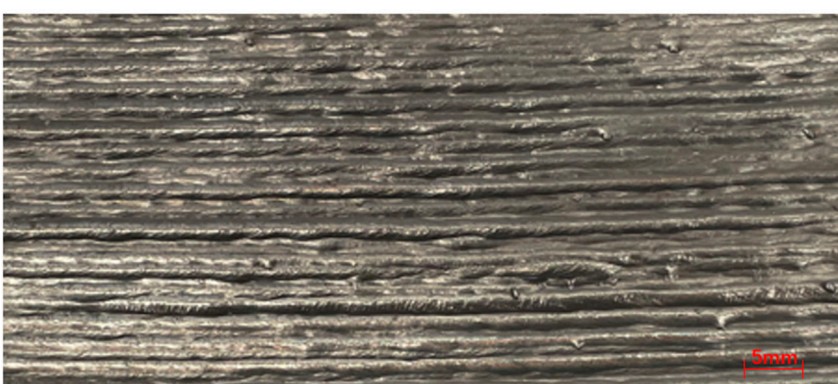

**Figure 5.** Surface formation of as-deposited wall.

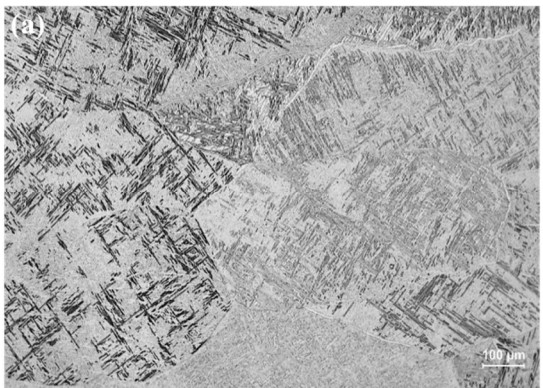 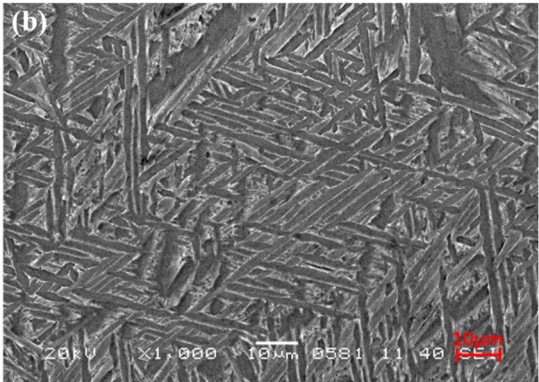

**Figure 6.** Microscopic structure of as-deposited wall: (**a**) optical microstructure: (**b**) SEM microstructure.

### 3.2. Role of Heat Treatment in Sample

Solution treatment followed by an aging heat treatment strategy is currently the most commonly used heat treatment strategy for additively manufactured Ti-6Al-4V alloy structural components. Figure 7 illustrates the growth process of the $\alpha$ phase in LAHAM Ti-6Al-4V alloy during the solution treatment process. During this process, the $\alpha$ phase undergoes continuous growth and intersection, causing fragmentation of the primary and secondary $\alpha$ grain and the formation of a denser basket-weave microstructure. The aspect ratio of the $\alpha$ phase significantly influences the mechanical properties of titanium alloys. Complete fragmentation of large primary $\alpha$ grains is crucial for optimizing the mechanical properties of LAHAM Ti-6Al-4V alloy samples. Smaller $\alpha$ grains result in a more uniform microstructure distribution, leading to enhanced overall mechanical properties, particularly in terms of plasticity and toughness.

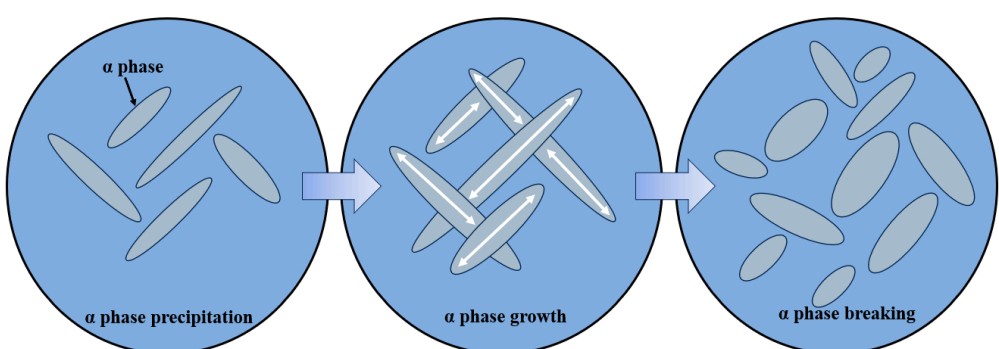

**Figure 7.** Schematic diagram of the growth process of the $\alpha$ phase in LAHAM Ti-6Al-4V alloy under heat treatment.

Given the similarities in the phases and grain sizes of LAHAM Ti-6Al-4V alloy under different heat-treatment conditions, this study conducted XRD analysis to characterize the samples in the 950 °C/1 h, AC + 540 °C/6 h, AC, and as-deposited states. Figure 8 presents the results of the XRD analysis, revealing significant peak broadening and the presence of the $\beta$ phase in the LAHAM Ti-6Al-4V alloy samples under the as-deposited state and the 950 °C/1 h, AC + 540 °C/6 h, AC heat-treatment conditions. In combination with the microstructure shown in Figure 6b, it is evident that for the as-deposited components, the quantity and size of the $\beta$ phase can be considered negligible. After solution treatment at 950 °C, a majority of the $\alpha'$ martensite in the components decomposes, forming an $\alpha + \beta$ structure. In Figure 8, the $\beta$ peak (110) of the samples under heat-treatment conditions appears at $2\theta = 38°$, with the $\beta$ peak under 950 °C/1 h, AC + 540 °C/6 h, AC heat-treatment conditions, the width of $\beta$ peak under heat-treatment conditions is slightly wider than that

of β phase peak under as-deposited conditions. The presence of the β phase in titanium alloys significantly enhances their plasticity, improving their deformability and ductility. Therefore, the plasticity of the LAHAM Ti-6Al-4V alloy components under heat-treatment conditions is notably higher than that of the components in the as-deposited state.

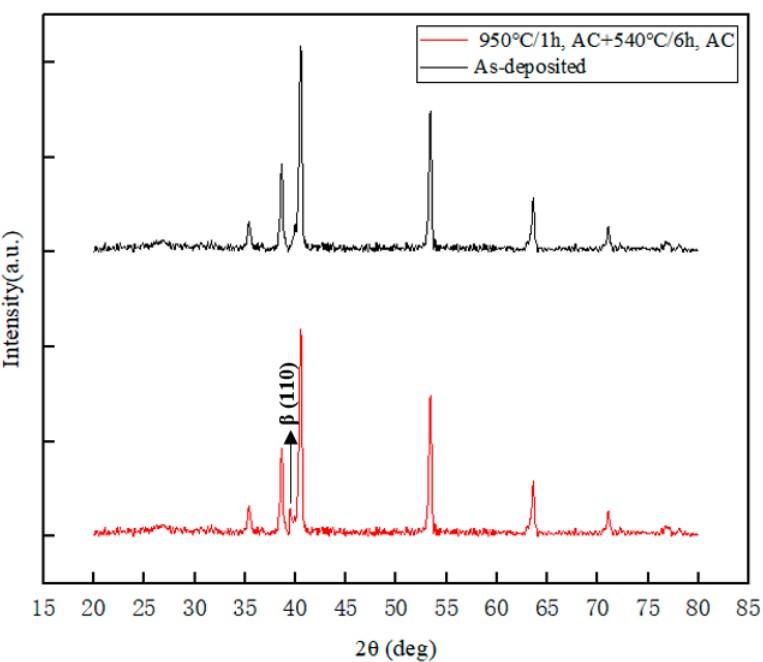

**Figure 8.** XRD analysis of LAHAM Ti-6Al-4V alloy under heat-treatment conditions.

*3.3. After Heat Treatment Microstructures*

3.3.1. Influence of Solution Temperature

Figure 9a–c present the optical microstructures of LAHAM Ti-6Al-4V alloy after 1 h solution treatment at 900 °C, 950 °C, and 1000 °C, followed by air cooling and 6 h of aging heat treatment at 540 °C. After solution treatment, the microstructure transitions from the typical Widmanstätten structure of the as-deposited state to a basket-weave structure, which consists predominantly of the primary α phase and transformation β phase, wherein the transformation β phase primarily comprises the secondary α phase and β phase [18]. The SEM micrographs of LAHAM Ti-6Al-4V alloy at different solution temperatures are illustrated in Figure 10a–c. From Figure 5, it is observed that as the solution treatment temperature increases from 900 °C, 950 °C to 1000 °C, the thickness of α lamellae increases, while the aspect ratio decreases.

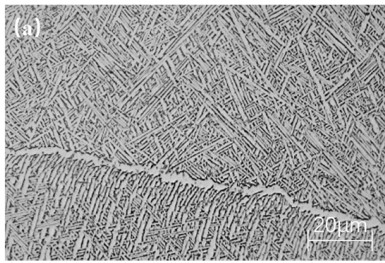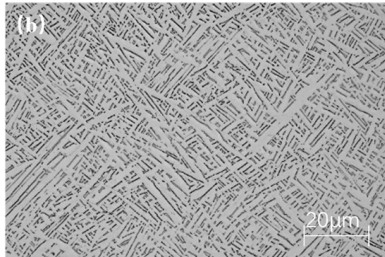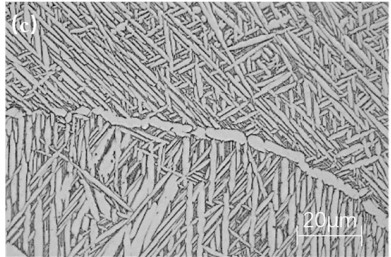

**Figure 9.** Optical microstructure under different solution treatment temperature conditions: (**a**) 900 °C/1 h, AC + 540 °C/6 h, AC; (**b**) 950 °C/1 h, AC + 540 °C/6 h, AC; (**c**)1000 °C/1 h, AC + 540 °C/6 h, AC.

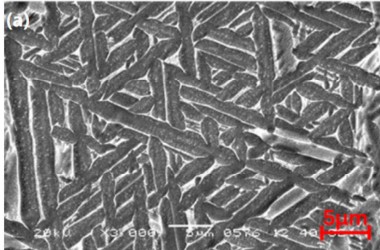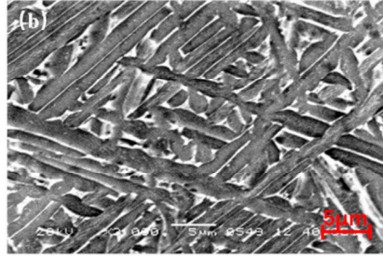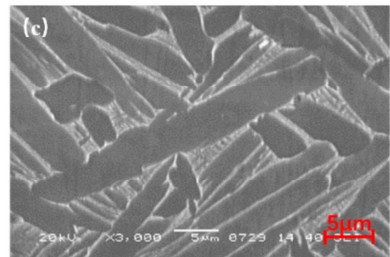

**Figure 10.** SEM microstructure under different solution treatment temperature conditions: (**a**) 900 °C/1 h, AC + 540 °C/6 h, AC; (**b**) 950 °C/1 h, AC + 540 °C/6 h, AC; (**c**) 1000 °C/1 h, AC + 540 °C/6 h, AC.

At 1000 °C, significant recrystallization occurs within β grains, and due to the precipitation of α phase at β grain boundaries, the previous β grains are no longer continuous. Additionally, Figure 9c presents the β grains; after solution treatment, these transform from columnar to equiaxed form. Figure 11 presents the distribution of α lamellar thicknesses under different heat treatments. The α lath thickness was quantified using Image J software (x64), indicating an increase in α lamellar thickness with the rise in the solution temperature.

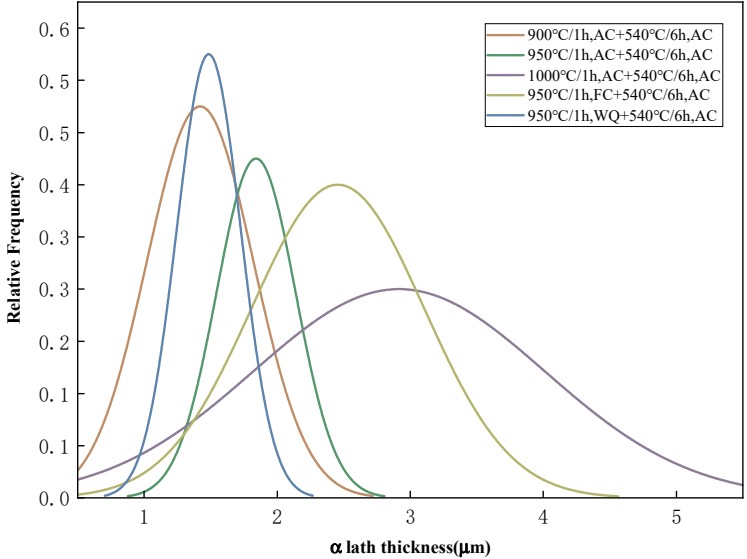

**Figure 11.** Distribution of alpha lamellar thickness under different heat treatment regimes.

### 3.3.2. Influence of Cooling Methods

Figure 12a–c display the optical microstructures of LAHAM Ti-6Al-4V alloy after furnace cooling, air cooling, and water quenching to room temperature following 1 h solution treatment at 950 °C, then aging at 540 °C for 6 h before air cooling. SEM microstructures under similar conditions are presented in Figure 13a–c. It is evident from Figures 11 and 12 that as the cooling rates increase, significant changes occur in the microstructure. The thickness of the original α lamellae gradually decreases, while the aspect ratio increases. As depicted in Figure 12a, the slower cooling rate during furnace cooling allows sufficient time for atomic diffusion, providing favorable conditions for the nucleation, growth, and coalescence of the secondary α phase. Consequently, the grain size under furnace-cooling conditions is notably larger than that under air and water-quenching conditions, with partial α lamellae exhibiting spheroidization under slower cooling rates, as shown in Figure 12b. Under air-cooling conditions, a typical basket-weave structure is obtained, wherein the high-temperature β phase transforms into secondary α phase

and β phase at an appropriate cooling rate, resulting in reduced α lamellae width compared to furnace-cooling conditions, with adjacent α lamellae oriented differently and interlaced, as illustrated in Figure 13b. Under water-quenching conditions, as observed in Figures 12c and 13c, the α lamellae thickness is significantly smaller than that obtained under the furnace and air-cooling conditions. Moreover, the presence of acicular martensite α′ phase with a large aspect ratio in Figure 12c indicates martensitic transformation post-solution treatment, as the rapid cooling rate prevents sufficient atomic diffusion, resulting in the generation of martensite α′ phase from high-temperature β phase under faster cooling conditions [19]. The reduction in α lamellae thickness with the increase in the cooling rate observed in the figures will impact the mechanical properties of the samples.

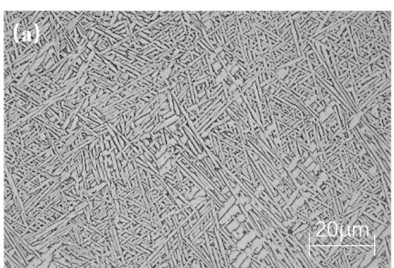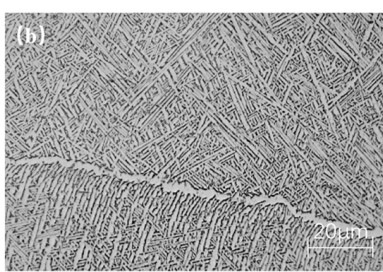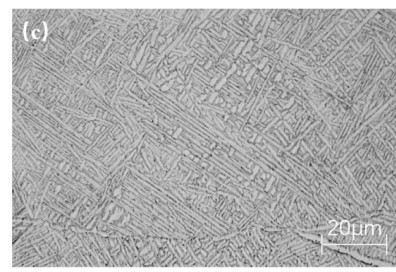

**Figure 12.** Optical microstructure under different cooling rate conditions: (**a**) FC; (**b**) AC; (**c**) WQ.

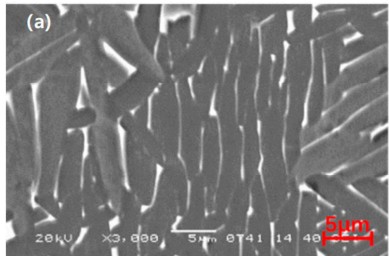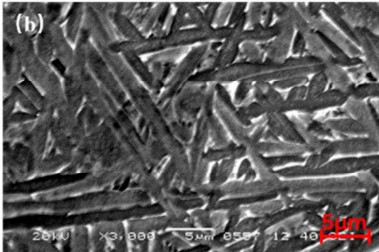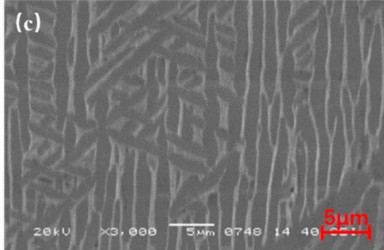

**Figure 13.** SEM microstructure under different cooling rate conditions: (**a**) FC; (**b**) AC; (**c**) WQ.

### *3.4. Mechanical Properties*

#### 3.4.1. As-Deposited Mechanical Properties

The performance of dual-phase titanium alloys is closely related to their chemical composition, phase chemistry, volume fraction, stability, strength, and microstructure [20]. The lower layers of the LAHAM Ti-6Al-4V alloy in the as-deposited state experience multiple thermal cycles, resulting in coarser grains and a quenched microstructure. The average tensile strength is 1100 MPa, the average impact toughness is 15.48 J/cm$^2$, and the average hardness is $360 \pm 25.3$ HV.

#### 3.4.2. Different Solution Treatment Temperatures

Figure 14 presents the tensile properties of the LAHAM Ti-6Al-4V alloy after it has been exposed to different solution treatment temperatures. The as-deposited LAHAM Ti-6Al-4V alloy exhibits higher ultimate tensile strength and lower ductility. Upon solution treatment, the specimens exhibit improved ductility, with their strength increasing as the solution treatment temperature rises from 900 °C to 950 °C. However, when the solution treatment temperature increases from 950 °C to 1000 °C, the strength decreases while the ductility continues to decline. Previous studies have shown that the aspect ratio of α laths significantly influences the variations in the tensile strength of Ti-6Al-4V alloys [21]. As the solution treatment temperature rises to below the β transit temperature, the aspect ratio of α laths decreases. Consequently, as more numerous and smaller secondary α phases form, the hindrance to crack propagation increases, leading to improvements in the

tensile strength [22]. However, at 1000 °C, the growth of α phases at β grain boundaries disrupts the continuity of the β grain boundaries, reducing their effectiveness in resisting deformation and resulting in decreased tensile strength and elongation. The plasticity of the material is limited by crack nucleation resistance and crack propagation resistance. Larger α laths, which are characteristic of α′ martensite, negatively impact plasticity. Therefore, at a solution treatment temperature of 950 °C, the specimen exhibits a balanced combination of tensile strength and elongation.

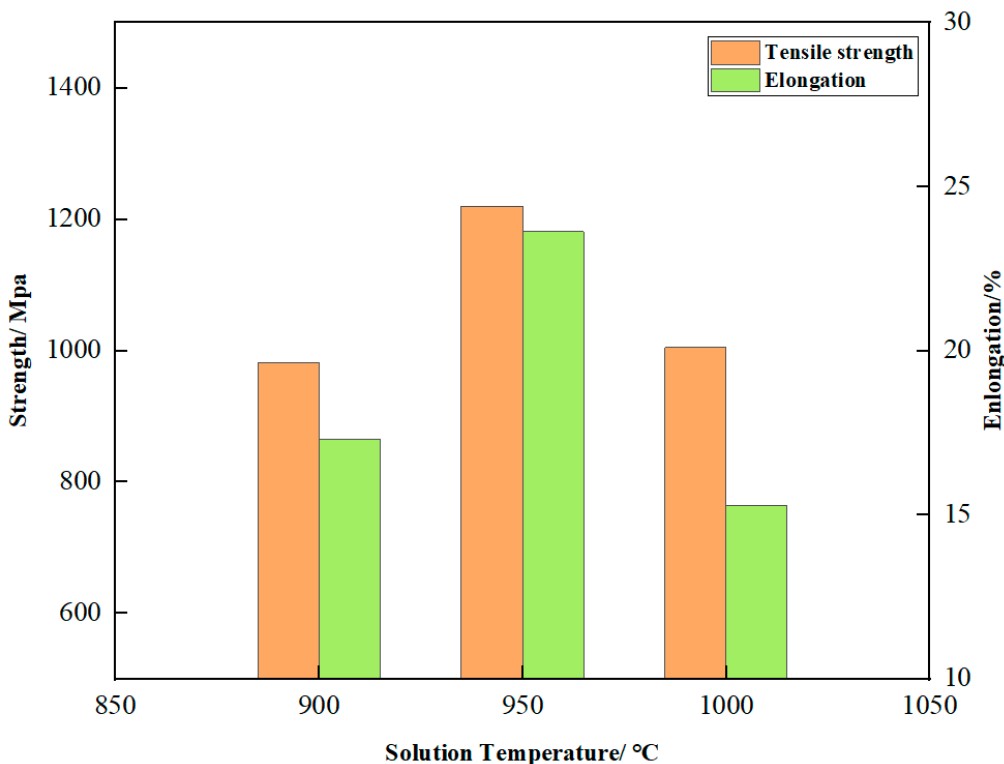

**Figure 14.** Tensile properties of LAHAM Ti-6Al-4V alloy after solution treatment at 900 °C, 950 °C, and 1000 °C.

In impact experiments, the energy absorbed during sample fracture includes both crack initiation and propagation energies, reflecting the material's ability to resist crack generation and extension [23]. Figure 15 illustrates the impact toughness of LAHAM Ti-6Al-4V alloy at different solution treatment temperatures, and also represents the energy absorbed by the specimen at the fracture points. Corresponding SEM images of impact fracture surfaces are depicted in Figure 16a–c. It is evident that the impact toughness of LAHAM Ti-6Al-4V alloy significantly improves after the solution treatment. This improvement is attributed to the transformation of coarse Widmanstätten structures into interlocking basket-weave structures, where larger orientation differences between adjacent α grains lead to increased resistance to crack propagation [24]. With increasing solution treatment temperature, impact toughness initially rises before declining. At 950 °C, the proliferation of small secondary α phases impedes crack propagation, enhancing impact toughness. However, at 1000 °C, the significant coarsening of the microstructure leads to reduced impact toughness.

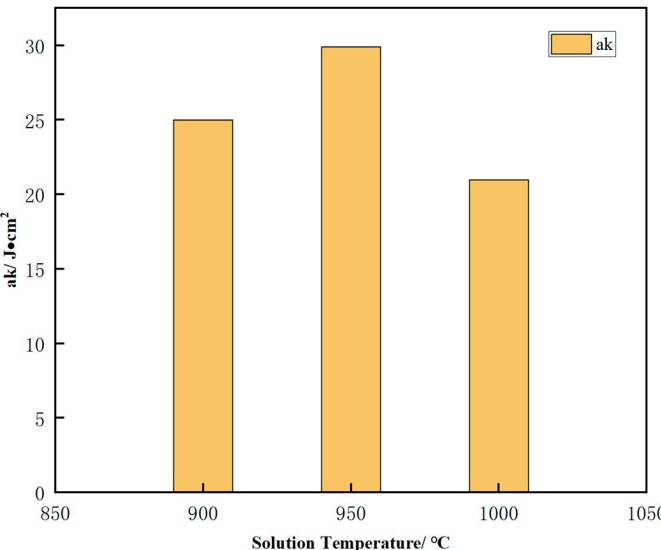

**Figure 15.** Impact toughness of LAHAM Ti-6Al-4V alloy after solution treatment at 900 °C, 950 °C, and 1000 °C.

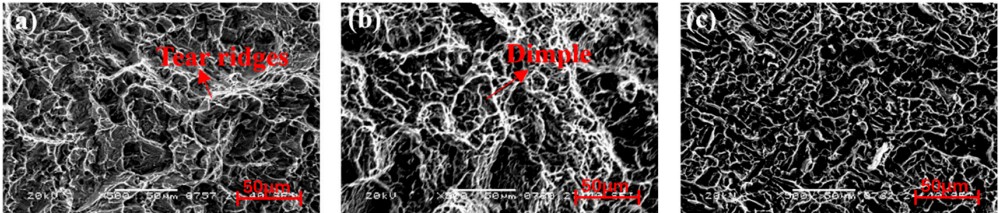

**Figure 16.** SEM morphology of impact fracture surfaces of LAHAM Ti-6Al-4V alloy at different solution treatment temperatures: (**a**) 900 °C; (**b**) 950 °C; (**c**) 1000 °C.

Figure 17 illustrates the hardness of LAHAM Ti-6Al-4V alloy specimens at different solution heat treatment temperatures. As the temperature of the solid solution increases, the hardness of the sample increases and its distribution becomes more uniform. This is attributed to the predominantly interlocked basket-weave structure of the specimens after solution treatment, which is more uniform and less quenched compared to that of the as-deposited Widmanstätten structures. Furthermore, the microhardness of LAHAM Ti-6Al-4V alloy decreases with the increase in the solution heat treatment temperature. At 950 °C, the complete decomposition of $\alpha'$ martensite results in reduced hardness, while at 1000 °C, the coarser $\alpha$ laths and increased grain boundary spacing lead to reduced hardness.

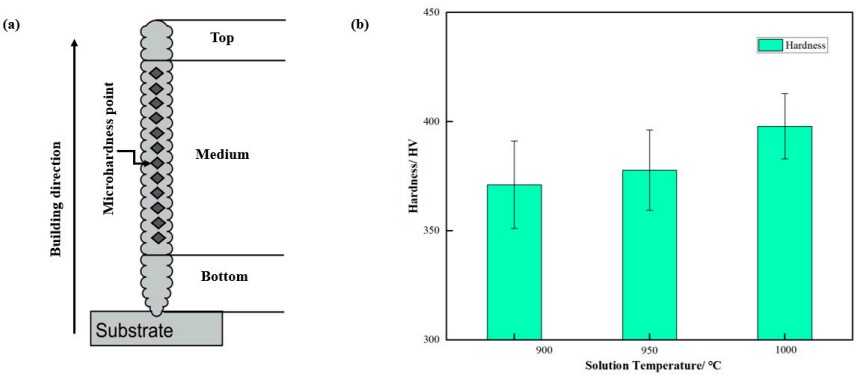

**Figure 17.** Microhardness measurement: (**a**) schematic diagram of microhardness position of sample; (**b**) microhardness of LAHAM Ti-6Al-4V alloy after solution treatment at 900 °C, 950 °C, and 1000 °C.

### 3.4.3. Different Solution Cooling Methods

Figure 18 presents the room temperature tensile properties of LAHAM Ti-6Al-4V alloy under different cooling rates. It is observed that the ultimate tensile strength of the specimens increases continuously with cooling rates, while the ductility initially increases before decreasing. The specimens subjected to air cooling after solution treatment at 950 °C for 1 h exhibit the best ductility. The grains obtained under furnace-cooling conditions are the coarsest. Larger grain sizes provide more active slip systems, thereby enhancing the ductility [25]. However, excessive grain size reduces the material's ability to deform coherently, leading to reduced ductility [26]. Consequently, the ductility of the specimens under furnace-cooling conditions is superior to that of those under water-cooling conditions but inferior to those under air-cooling conditions. When the grain size is larger, fewer grain boundaries are present to hinder slip. Thus, larger grains have a detrimental effect on strength. The presence of martensite in the water-cooled specimens, with numerous dislocations and twin boundaries, enhances strength but reduces ductility after aging [27]. The $\alpha$ laths in the air-cooled specimens are significantly smaller than those in the furnace-cooled specimens. A smaller grain size enhances the material's ability to resist deformation, resulting in a superior strength–ductility balance in the air-cooled specimens.

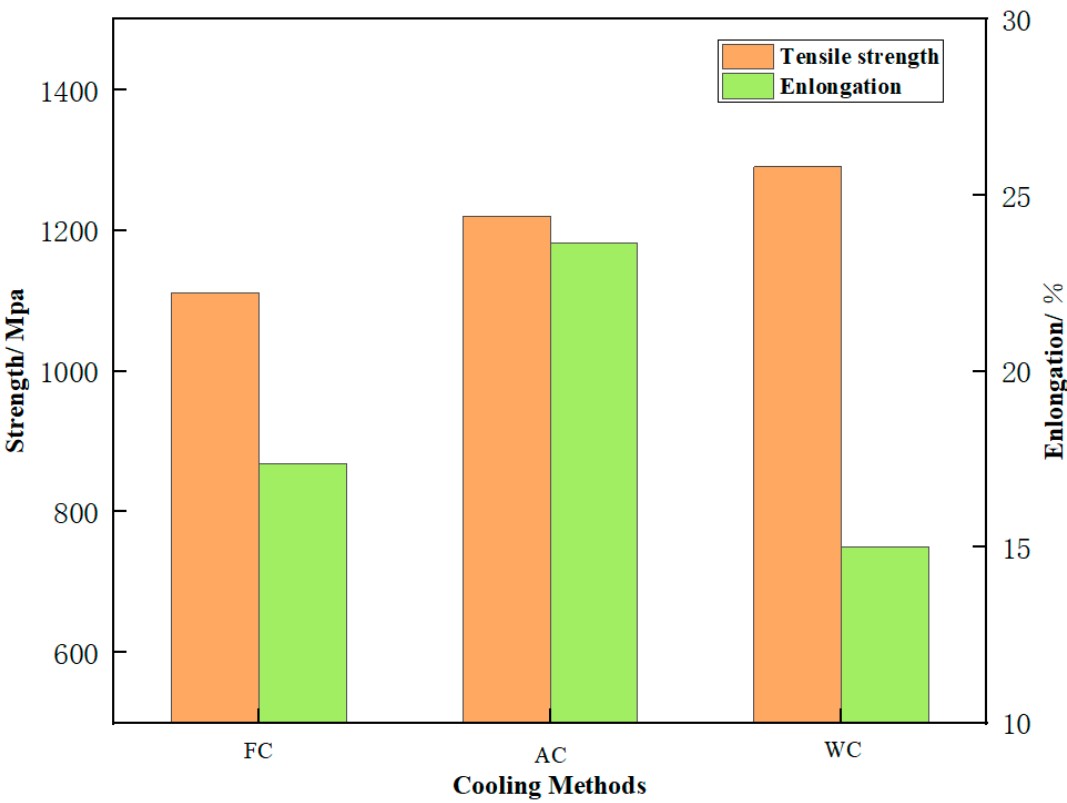

**Figure 18.** Tensile properties of LAHAM Ti-6Al-4V alloy under FC, AC, and WQ conditions.

Figure 19 presents the impact toughness of LAHAM Ti-6Al-4V alloy subjected to different cooling methods at room temperature. SEM fracture morphology of specimens under different cooling rates is presented in Figure 20a–c. As the cooling rate increases, the impact toughness of the specimens initially increases before decreasing, with the poorest impact toughness observed under water-cooling conditions. Fractures under water-cooling conditions exhibit numerous brittle fracture edges and cleavage steps, indicating poor impact toughness. In contrast, fractures under furnace and air-cooling conditions exhibit numerous irregular dimples, consistent with better toughness of these specimens. From a microstructural perspective, it is reported that during fracture, the $\alpha$ phase serves as the channel for crack generation and propagation in dual-phase titanium alloys, with toughness

increasing as the average free path length within the α phase [25,28–31]. The formula for calculating the average free path length within the α phase is provided in Equation (1). Microstructural analysis reveals that the α lath thickness is minimal under water-cooling conditions, with a significant presence of α′ martensite, resulting in the lowest impact toughness. The impact toughness of specimens obtained under furnace-cooling conditions is higher than those obtained under water-cooling conditions but lower than those obtained under air-cooling conditions. This is primarily due to the larger grain size obtained under furnace-cooling conditions, which, according to Equation (1), favors increased impact toughness due to thicker lamellar layers.

$$L_m = \left(\frac{4T_0}{3f}\right)(1-f) \tag{1}$$

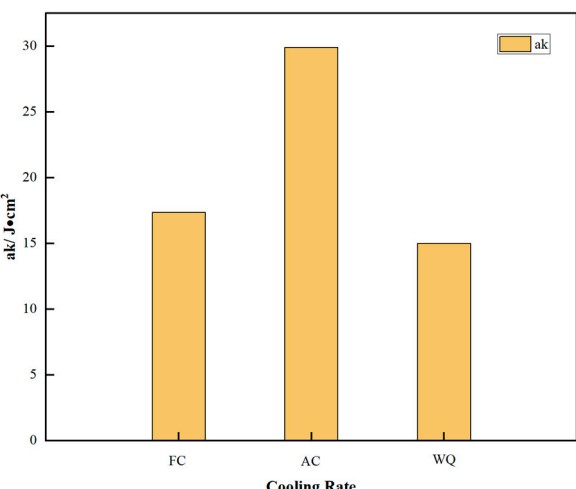

**Figure 19.** Impact toughness of LAHAM Ti-6Al-4V alloy under FC, AC, and WQ conditions.

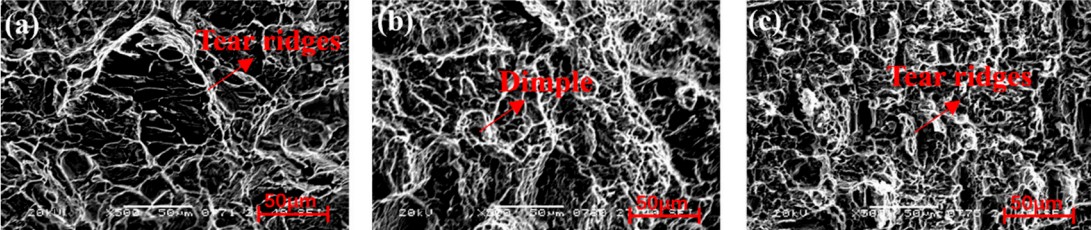

**Figure 20.** SEM morphology of impact fracture surfaces of LAHAM Ti-6Al-4V alloy under different cooling rates (**a**) FC; (**b**) AC; (**c**) WQ.

However, a larger grain size also reduces the material's ability to deform coherently, resulting in intermediate impact toughness [32]. Under air-cooling conditions, the smaller size of α and β phases results in their interlocking arrangement, with β grain boundaries hindering crack propagation.

Figure 21 depicts microhardness of LAHAM Ti-6Al-4V alloy after different cooling rates. It is observed that the hardness increases with faster cooling rates. The grains obtained under furnace-cooling conditions are coarser, with larger intergranular gaps and greater grain slip freedom, resulting in lower hardness. In contrast, the grains obtained under air-cooling conditions are smaller than those under furnace cooling. Conversely, the microstructure obtained under air-cooling conditions has a smaller grain size compared to the furnace-cooling conditions, with a higher number of grains per unit volume leading to reduced sliding freedom and increased hardness. The highest hardness is achieved under water-quenching conditions due to the martensitic transformation during the solid solution

water-quenching process. Martensite contains numerous dislocations that strongly pin the grain boundaries, resulting in the highest hardness being observed in the microstructure under water-quenching conditions.

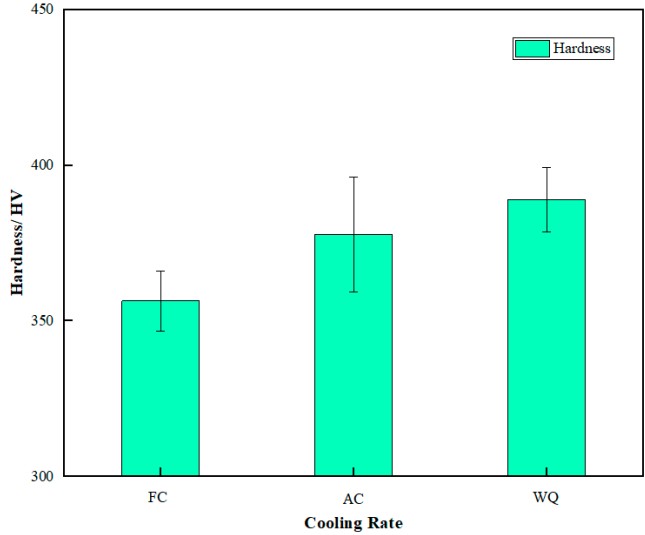

**Figure 21.** Hardness of LAHAM Ti-6Al-4V alloy under FC, AC, and WQ conditions.

## 4. Conclusions

This study focused on investigating the microstructural characteristics of LAHAM Ti-6Al-4V alloy after heat treatment. The effects of solution treatment temperature and cooling methods on the microstructure and mechanical properties of the LAHAM Ti-6Al-4V alloy were systematically analyzed.

(1) The microstructural characteristics of LAHAM Ti-6Al-4V alloy consist of columnar β grains with an average grain thickness of 300 μm, which grow epitaxially along the deposition direction. These grains primarily comprise Widmanstätten α laths and a small amount of acicular martensite α′.

(2) As the solution treatment temperature increases, the thickness of primary α laths increases, the aspect ratio of α laths decreases, and the volume fraction of secondary α phases increases. These changes lead to increased strength, decreased ductility, improved impact toughness, and decreased microhardness of the specimens. However, when the solution treatment temperature reaches 1000 °C, severe coarsening of α laths occurs, resulting in simultaneous decreases in strength and ductility. The impact fracture at this temperature exhibits brittle characteristics. The specimens exhibit favorable comprehensive mechanical properties at a solution treatment temperature of 950 °C.

(3) Upon cooling at a sufficiently slow rate, the high-temperature β phase transforms into the low-temperature α phase. At a moderate cooling rate, the high-temperature β phase transforms into plate-like α and β phases. With excessively rapid cooling, the high-temperature β phase undergoes a martensitic transformation, generating martensitic α′ phases. As the cooling rate increases, the strength and microhardness of the specimens continuously increase before decreasing. Under water quench conditions, the impact fracture exhibits brittle characteristics. The specimens demonstrate optimal mechanical properties under air-cooling conditions after solution treatment.

(4) During the solution treatment process, α grains grow, intersect with each other, and undergo significant fragmentation, resulting in a noticeable size reduction. Furthermore, the β phase in the LAHAM Ti-6Al-4V alloy increases significantly after solution treatment followed by aging, leading to a notable enhancement in the comprehensive mechanical properties of the samples after heat treatment compared to the components in the as-deposited state. To achieve optimal comprehensive mechanical properties for LAHAM Ti-6Al-4V alloy, a heat-treatment scheme of 950 °C/1 h, AC + 540 °C/6 h, AC is recommended.



**Author Contributions:** Resources, J.F., L.Z., G.C. and Y.Q.; Writing—original draft, Y.C.; Writing—review & editing, Y.Z. and F.W. All authors have read and agreed to the published version of the manuscript.

**Funding:** This research received no external funding.

**Institutional Review Board Statement:** Not applicable.

**Informed Consent Statement:** Not applicable.

**Data Availability Statement:** Data are contained within the article.

**Conflicts of Interest:** Author Lilong Zhou was employed by the company Aerospace Engineering Equipment (Suzhou) Co., Ltd.; Authors Guoqiang Chen and Yonghui Qin were employed by the company Jiangsu Yangzi-Mitsui Shipbuilding Co., Ltd. The remaining authors declare that the research was conducted in the absence of any commercial or financial relationships that could be construed as a potential conflict of interest.

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
