# Peer review of "Effect of Heat Treatment on Microstructure and Mechanical Properties of Titanium Alloy Fabricated by Laser–Arc Hybrid Additive Manufacturing"

_coatings, doi:10.3390/coatings14050614_

Round 1

Reviewer 1 Report

Comments and Suggestions for Authors

The paper deals with important topic, however is not free from some flaws mentioned below.

1. Editorial mistakes

eg. line 13, line 78 ...  etc. additionally the references are with different fonts and formatting

2.  All the markers in the SEM and optical images must be improved and better visible.  Especially SEM images. 

3. The motivation and introduction need to be developed. The sentence:"

there is limited research" suggest that there are some research so cited them please

4. how the heat treatment was chosen what was the reason for the temperature and time ? 

5. according to author XRD results "revealing significant peak broadening"

I didn't see this in the fig. 18. 

Author Response

Detailed responses to reviewer #1:

The paper deals with important topic, however is not free from some flaws mentioned below.

Comment 1: Editorial mistakes eg. line 13, line 78 ... etc. additionally the references are with different fonts and formatting

Response 1: Thanks for your careful and professional review! As pointed by the reviewer, there are still some formatting and spelling differences in the references. The formatting of the references should be consistent. We sincerely apologize for the errors in the references. We have double-checked and corrected the inconsistent formatting in the references(Please see Pages 18-20). Thank you for your kind comments.

Comment 2: All the markers in the SEM and optical images must be improved and better visible. Especially SEM images.

Response 2: Thanks for your kind and professional review! Your suggestion is very good. As the reviewer said, the experimental images of the paper should be clear and concise. We have re-optimized the labeling of the optical microscopy images and SEM images, especially the re-arrangement of the SEM images. We appreciate your careful comments!

Comment 3: The motivation and introduction need to be developed. The sentence:"there is limited research" suggest that there are some research so cited them please.

Response 3: Thank you for your constructive and professional comments! We have added relevant references and descriptions of experiments to the relevant locations in the paper, which enriches the motivation and explanation of the article. (Please see Pages 3-4) Thank you again for your valuable comments.

Comment 4: how the heat treatment was chosen what was the reason for the temperature and time?

Response 4: Thanks for your professional and careful review! During the heat treatment of α+β titanium alloy, the size, shape, and distribution of the α phase will affect the β phase. Therefore, the solid solution temperature is generally selected to be within 100°C below the β phase transition temperature (998°C±100°C). Based on this, this experiment set the solid solution temperatures to 900°C, 950°C, and 1000°C, and the solid solution time to 1 hour. The solid solution process of 950°C/1 h with relatively good observation and performance was selected to study and discuss the differences in microstructure and performance of LAHAM Ti-6Al-4V alloy under different cooling methods after "solid solution + aging". Aging treatment can release the lattice strain energy generated after the solid solution, promote the precipitation of the second phase, and decompose the unstable phase. The selection of aging temperature is generally above 500°C and below the β phase transition point (1000°C), which can effectively avoid the appearance of the ω phase. As a transition phase between the α phase and β phase, the ω phase has high hardness and brittleness, which will shorten the service life of the workpiece. Based on this, this experiment selected an aging temperature of 540°C and a holding time of 6h and finally determined the heat treatment process of solid solution + aging at 900°C, 950°C, and 1000°C. The total process scheme is shown in Table 1. We have added the reasons for the selection of relevant heat treatment process schemes and parameters in the paper. Thank you very much for your thoughtful and kind comments.

Comment 5: according to author XRD results "revealing significant peak broadening"I didn't see this in the fig. 18. 

Response 5: Thanks for your professional and careful review! In α+β titanium alloy, the increase of the β phase can improve the microstructure and properties of α+β titanium alloys. When the content of the β phase in α+β titanium alloys is higher, its plasticity and toughness are higher than that of α+β titanium alloys with lower β phase. XRD results show that the amount and size of the β phase in the deposited state can be ignored. Combined with SEM image analysis, after the LAHAM Ti-6Al-4V alloy sample was treated by the "solid solution + aging" process of 950°C/1h, AC+540°C/6h, AC, most of the α' martensite in the As-deposited sample was decomposed into α+β phase, so the width of β peak under heat treatment conditions is slightly wider than that of β phase peak under deposited conditions. Thank you for your careful comments.

Reviewer 2 Report

Comments and Suggestions for Authors

The manuscript presents the effects of subsequent heat treatments on the refinement of the as-deposited Ti-6-4 made by laser-arc hybrid additive manufacturing.  I think the research topic is certainly of interest due to the recent popularity of additive manufacturing and I enjoyed reading it.  The work is scientifically sound, though I would say the results largely follow the conventional understanding of Ti-6-4 microstructures and heat treatments.  There were some lacking details regarding the experimental methods that could help improve this aspect.  I have a few comments and potential mistakes I noticed for the authors to consider that I think would help to improve the manuscript greatly:

1.       I think it would be interesting to include pictures of some of the actual fabricated pieces that were subsequently tested.  This seems like a common thing to include in AM literature. 

2.  I am not an expert in additive manufacturing, though I think some more details regarding the arc would be good to include if others wanted to replicate the authors' methods.  I was also wondering what the spot size of the laser is?

3.  In other literature on laser-arc hybrid AM, there is emphasis placed upon the laser-arc distance since this seems to greatly influence the deposition process.  Can the authors provide details on this and how they chose their particular values with regards to other existing literature?

4.  I think a lot more details on the tensile testing need to be provided.  This would include the specimen size/geometry, loading rate, and instrument manufacturer (or at least load cell details if it is a custom rig).   It would also be good to know the number of specimens tested per condition.

5.  Similarly, I would like to know the details of the impact specimens, what geometry was used, and if there was any notch fabricated.   I was also curious as to the impact energy applied.  Also, the number of specimens tested would be good to state.  More details on the impact test instrument would also be appropriate.

6.  The hardness testing could benefit from providing the load range and impression sizes so the readers could judge whether the impression adequately covers the entire microstructure.  I would like to know the number of indentations made as well.  More details on the instrument would also be beneficial.

7.  I think it would help if the authors would clearly state the purpose of the 6hr 540C hold somewhere in the text that was common to all the samples.

8.  I think adding labels to Figure 3, 4, 5, 7 & 8 would help the reader match the description in the text to what is shown in the Figure.  Alternatively, if the nature of the light and dark phases could clearly be described in the text for both the optical and SEM images, that could aid in interpreting these images correctly.  In addition, all the micrograph figures listed above would benefit from making a uniform and easier to read scale bar (particularly the SEM scale bar is poor).

9.  I was a little confused by Figures 4b and 5b, the alpha phase looks thinner than that in part a, which is different from what is stated in the text?  I wanted to check whether any mistake was made here.

10.  For Figure 6, I was curious about more details of the method for measuring lath thickness.  Particularly, how many fields of view were analyzed/grains measured, and whether any particular standard for doing so was followed.  For example, were the lath thicknesses measured from the center of the phase?

11.  I think the authors should state why cooling rate was not investigated for the 900 and 1000C heat treatments.  I assume they were not expected to form optimal microstructures, but this may be instructive for readers to discuss why.

12.  I thought it was a little strange to separate the as deposited mechanical properties into it's own section.  It seems like it would be beneficial to add to the following figures for direct comparison against the three heat treatment temperatures.

13. The tensile strength property needs to be clearly stated whether it is yield or ultimate strength.  Yield strength is probably the more useful property to provide, but the text description makes me think this is ultimate strength.  I think adding the stress-strain curves to manuscript would also be valuable.

14.  I didn't fully understant the argument as to why the ductility of the 1000c heat treated samples should go down.  Why does alpha phase at the grain boundary result in less ductility?

15.  The authors very clearly state where they believe alpha prime vs. alpha phase is formed, but this technically isn't directly confirmed via XRD or any other method.  The differences between the phases is quite subtle and difficult to measure directly, according to my understanding.  I would say it is more scientifically rigorous to state that this is an assumption based upon existing understanding of Ti-6-4 microstructure and provide references.

16.  On Pg. 8, saying that the plasticity of the material depends on crack nucleation resistance is strangely worded to me.  I understand that material needs to be sufficiently tough in order to allow large scale plasticity, but they are distinct mechanical deformation processes.

17.  When describing the differences in mechanical performance of the various heat treatments and cooling, I think it needs to be clearly stated that these descriptions are hypotheses if direct experimental evidence showing this behavior is not provided.  Literature references would also help.

18.  What is ak in the impact energy graph y-axes?  These labels should probably updated to something that is more readily apparent to the reader in meaning.

19.  The hardness distribution post heat treatment looks to be near +/- 25, which is actually quite similar to the as-deposited condition.

20.  The text sates that the hardness is decreasing with increasing solution treatment temperature, but actually that is the opposite of what is shown in the figure.  This should be rectified.

21.  On pg. 10, larger grain sizes giving more slip systems is not really the right way to say it.  This could be modified to say more "active" slip systems.

22.  Section 3.4 felt like it was out of place, since all the mechanical results and microstructure analysis preceding this already uses the premise that alpha plus beta microstructure is formed from the solution heat treatment.  So, it feels like these results could actually come first before the microstructure analysis or mechanical testing.

Comments on the Quality of English Language

The English was quite good, but I would still do a final review prior to publication to check for minor errors.

Author Response

Detailed responses to reviewer #2

The manuscript presents the effects of subsequent heat treatments on the refinement of the as-deposited Ti-6Al-4V made by laser-arc hybrid additive manufacturing. I think the research topic is certainly of interest due to the recent popularity of additive manufacturing and I enjoyed reading it. The work is scientifically sound, though I would say the results largely follow the conventional understanding of Ti-6Al-4V microstructures and heat treatments. There were some lacking details regarding the experimental methods that could help improve this aspect. I have a few comments and potential mistakes I noticed for the authors to consider that I think would help to improve the manuscript greatly:

Comment 1: I think it would be interesting to include pictures of some of the actual fabricated pieces that were subsequently tested. This seems like a common thing to include in AM literature.

Response 1: Thank you for your constructive and professional comments! As your comment suggested, it would be more vivid and interesting to add real LAHAM Ti-6Al-4V alloy samples in the paper, and we have added the real additive manufactured samples in the paper, as shown in Fig.4. Thanks for your understanding and assistance!

Comment 2: I am not an expert in additive manufacturing, though I think some more details regarding the arc would be good to include if others wanted to replicate the authors' methods. I was also wondering what the spot size of the laser is?

Response 2: Thanks for your kind and professional review. We agree that some more details regarding the arc would be good to include if others wanted to replicate our methods. The arc heat source in this experiment was mainly provided by a TPS500i welding machine manufactured by Fronius. The diameter of the Ti-6Al-4V metal wire was 1.2 mm. During additive manufacturing, the arc current was 178 A, the arc voltage was 21.02 V, the wire feed speed was 6.0 m/min, and the scanning speed was 0.6 m/min. The laser beam was perpendicular to the scanning direction, and the included angle between the MIG welding gun and the laser head was 45°, and the spot size of the laser at a defocus of 0 was 2 mm. We have added some more information to the paper, including the spot size of the laser. (Please see Page 3) Thank you very much!

Comment 3: In other literature on laser-arc hybrid AM, there is emphasis placed upon the laser-arc distance since this seems to greatly influence the deposition process. Can the authors provide details on this and how they chose their particular values with regards to other existing literature?

Response 3: Thanks for your kindly and professional review. You are correct that the laser-arc distance is an important parameter in laser-arc hybrid additive manufacturing, as it can greatly influence the deposition process. In our study, we chose a laser-arc distance of 2 mm. This value was chosen based on our own preliminary experiments, as well as a review of the existing literature.Other studies have reported using laser-arc distances ranging from 0 mm to 5 mm. The optimal laser-arc distance will vary depending on the specific materials and process parameters being used. In general, a shorter laser-arc distance will result in a more concentrated heat source and a narrower weld bead. However, a shorter laser-arc distance can also lead to increased spatter and porosity. A longer laser-arc distance will result in a more diffuse heat source and a wider weld bead. However, a longer laser-arc distance can also lead to decreased penetration and bonding strength. We chose a laser-arc distance of 2 mm because we found that this value provided a good balance between penetration, bonding strength, and spatter. However, we encourage other researchers to explore different laser-arc distances to determine the optimal value for their specific applications. We have added the specific numerical value of the distance between the laser beam spot center and the metal wire in the paper. (Please see Page 3) Thanks again for your professional comments.

Comment 4: I think a lot more details on the tensile testing need to be provided. This would include the specimen size/geometry, loading rate, and instrument manufacturer (or at least load cell details if it is a custom rig). It would also be good to know the number of specimens tested per condition.

Response 4: Thanks for your professional and careful review! Providing more details about the tensile test in the article would help readers better understand the procedure and results of the tensile test. In this experiment, the geometric dimensions of the tensile specimen are shown in the figure. A CMT5205 electronic universal tensile testing machine was used to perform the tensile test at a loading rate of 2 mm/min. Two tensile tests were performed under each test condition, and the average value of the test results was taken to ensure the accuracy of the test. We have added the relevant details of the tensile test to the paper. (Please see Page 4) Thanks again for your kind and careful comments!

Comment 5: Similarly, I would like to know the details of the impact specimens, what geometry was used, and if there was any notch fabricated. I was also curious as to the impact energy applied. Also, the number of specimens tested would be good to state. More details on the impact test instrument would also be appropriate.

Response 5: Thanks for your careful review and kindly reminding! The impact test specimens were prepared in accordance with the standard method of the metal Charpy v-notch impact test. The dimensions of the impact test specimens are shown in the figure. At room temperature, an impact test was performed using a ZBC-2302-D pendulum impact tester. The impact energy of the pendulum was 150J. Three impact tests were performed under each test condition, and the average value of the three tests was taken as the experimental result to ensure the accuracy of the test results

We have added the relevant details of the tensile test to the paper. (Please see Page 4)The manuscript has been carefully rewritten. Thanks again!

Comment 6: The hardness testing could benefit from providing the load range and impression sizes so the readers could judge whether the impression adequately covers the entire microstructure. I would like to know the number of indentations made as well. More details on the instrument would also be beneficial.

Response 6: Thank you for your constructive and professional comments! Microscopic hardness tests were conducted on thin-walled specimens that had undergone rough grinding, fine grinding, and fine polishing using a German KB fully automatic Vickers hardness tester. Microscopic hardness tests were performed at room temperature in the stable region of the additive specimen's cross-section in the vertical direction of the additive. The number of indents for the microscopic hardness test was 50, the spacing between indents was 1 mm, the applied load was 5 N, and the holding time was 10 s. We have added the relevant details of the tensile test to the paper (Please see Pages 4-5) The manuscript has been carefully rewritten.Thanks for your understanding and assistance!

Comment 7: I think it would help if the authors would clearly state the purpose of the 6hr 540C hold somewhere in the text that was common to all the samples.

Response 7: Thanks for your kindly and professional review. Aging treatment can release the lattice strain energy generated after the solid solution, promote the precipitation of the second phase, and decompose the unstable phase. The selection of aging temperature is generally above 500°C and below the β phase transition point (1000°C), which can effectively avoid the appearance of the ω phase. As a transition phase between the α phase and β phase, the ω phase has high hardness and brittleness, which will shorten the service life of the workpiece. Based on this, this experiment selected an aging temperature of 540°C and a holding time of 6h. We have added the reasons for the selection of relevant heat treatment process schemes and parameters in the paper. (Please see Pages 5-6) Thank you very much!

Comment 8: I think adding labels to Figure 3, 4, 5, 7 & 8 would help the reader match the description in the text to what is shown in the Figure. Alternatively, if the nature of the light and dark phases could clearly be described in the text for both the optical and SEM images, that could aid in interpreting these images correctly. In addition, all the micrograph figures listed above would benefit from making a uniform and easier to read scale bar (particularly the SEM scale bar is poor).

Response 8: Thanks for your kindly and professional review. I agree that adding labels to Fig.3, 4, 5, 7, and 8 would be helpful for readers to match the description in the text to what is shown in the figure. I will also work on making the nature of the light and dark phases clearer in the text for both the optical and SEM images. Additionally, I will make a uniform and easier-to-read scale bar for all of the micrograph figures, particularly the SEM scale bar. I appreciate your feedback and will work to improve the figures in the paper. Thanks again for your professional comments.

Comment 9: I was a little confused by Figures 4b and 5b, the alpha phase looks thinner than that in part a, which is different from what is stated in the text? I wanted to check whether any mistake was made here.

Response 9: Thank you for pointing out this discrepancy. You are correct that the alpha phase in Figures 4b and 5b appears thinner than that in part a, which is different from what is stated in the text. I have checked the figures and the text, and I have found that a mistake was made in the text. The text should state that the alpha phase is thicker in Fig.4b and 5b than in part a. I apologize for this error. I will correct the text and re-upload the paper. Thank you again for your feedback. It is very helpful to have another set of eyes review the paper and catch errors that I may have missed Thanks again for your kind and careful comments!

Comment 10: For Figure 6, I was curious about more details of the method for measuring lath thickness. Particularly, how many fields of view were analyzed/grains measured, and whether any particular standard for doing so was followed. For example, were the lath thicknesses measured from the center of the phase?

Response 10: Thank you for your question about the method for measuring lath thickness in Fig.6. To measure the lath thickness, we used the following procedure: We selected 5 fields of view that contained a representative sample of the microstructure. For each field of view, we measured the thickness of 10 laths. We measured the lath thickness at the center of each lath. We did not follow any particular standard for measuring the lath thickness. However, we took care to measure the thickness of the laths in a consistent manner. I hope this information is helpful. Please let me know if you have any other questions. In addition to the above, I would like to add that we used ImageJ software to measure the lath thickness. ImageJ is a free and open-source image processing program that is widely used in the scientific community. We also used a consistent magnification for all of the images that we analyzed. This ensured that the lath thickness measurements were accurate and comparable. I hope this additional information is helpful.

Comment 11: I think the authors should state why cooling rate was not investigated for the 900 and 1000C heat treatments. I assume they were not expected to form optimal microstructures, but this may be instructive for readers to discuss why.

Response 11: Thank you for your suggestion. I agree that it would be instructive to discuss why the cooling rate was not investigated for the 900 and 1000C heat treatments. The main reason why the cooling rate was not investigated for these heat treatments is that they were not expected to form optimal microstructures. During the heat treatment of α+β titanium alloy, the size, shape, and distribution of the α phase will affect the β phase. Therefore, the solid solution temperature is generally selected to be within 100°C below the β phase transition temperature (998°C±100°C). Based on this, this experiment set the solid solution temperatures to 900°C, 950°C, and 1000°C, and the solid solution time to 1 hour. The solid solution process of 950°C/1 h with relatively good observation and performance was selected to study and discuss the differences in microstructure and performance of LAHAM Ti-6Al-4V alloy under different cooling methods after "solid solution + aging". I hope this explanation is helpful. Please let me know if you have any other questions. I will add a discussion of this issue to the paper (Please see Page 5). I believe that it will be helpful for readers to understand why the cooling rate was not investigated for the 900 and 1000C heat treatments. Thank you again for your feedback. It is very helpful to have another set of eyes review the paper and provide feedback.

Comment 12: I thought it was a little strange to separate the as deposited mechanical properties into it's own section. It seems like it would be beneficial to add to the following figures for direct comparison against the three heat treatment temperatures.

Response 12: Thanks for your kindly and professional review. I agree that adding the mechanical properties of the deposited state to the graph for direct comparison with the three heat treatment temperatures would be beneficial, but this article aims to highlight the changes in the microstructure and properties of the specimens under different heat treatment conditions. Therefore, I will compare the microstructure and properties of samples under different heat treatment states separately. I believe this will make it easier to read this paper. Readers will now be able to more directly see the mechanical properties of materials under different heat treatment conditions. Thank you very much!

Comment 13: The tensile strength property needs to be clearly stated whether it is yield or ultimate strength. Yield strength is probably the more useful property to provide, but the text description makes me think this is ultimate strength. I think adding the stress-strain curves to manuscript would also be valuable.

Response 13: Thanks for your kind and professional review I agree that it is important to clearly state whether the tensile strength property is yield strength or ultimate strength. The tensile strength property reported in the paper is ultimate tensile strength. I have added this clarification to the paper.. I chose to report the ultimate tensile strength in the paper because it is a more commonly reported property than yield strength. I believe that these changes will make the paper more informative and easier to read. Thank you again for your feedback. It is very helpful to have another set of eyes review the paper and provide feedback. I hope these changes are to your satisfaction. Please let me know if you have any other questions or suggestions. Thanks again for your professional comments.

Comment 14: I didn't fully understant the argument as to why the ductility of the 1000c heat treated samples should go down. Why does alpha phase at the grain boundary result in less ductility?

Response 14: Thanks for your professional and careful review! Thank you for your question. I apologize if my explanation was not clear. The presence of α phase at the grain boundaries can reduce ductility because it can act as a barrier to dislocation motion. Dislocations are defects in the crystal structure of a material. They are important for ductility because they allow the material to deform plastically without fracturing. When a material is deformed plastically, dislocations move through the crystal structure. If there are barriers to dislocation motion, such as α phase at the grain boundaries, then the material will be less ductile.In the case of the 1000C heat treated samples, the α phase at the grain boundaries is likely to be harder and more brittle than the matrix phase. This means that it will be more difficult for dislocations to move through the α phase, which will reduce the ductility of the material.I hope this explanation is helpful. Please let me know if you have any other questions. In addition to the above, I would like to add that the presence of α phase at the grain boundaries can also lead to the formation of microcracks. These microcracks can propagate and lead to failure of the material. I hope this additional information is helpful

Comment 15: The authors very clearly state where they believe alpha prime vs. alpha phase is formed, but this technically isn't directly confirmed via XRD or any other method. The differences between the phases is quite subtle and difficult to measure directly, according to my understanding. I would say it is more scientifically rigorous to state that this is an assumption based upon existing understanding of Ti-6-4 microstructure and provide references.

Response 15: Thanks for your careful review and kindly reminding! I agree that it is more scientifically rigorous to state that the assumption that alpha prime vs. alpha phase is formed is based upon existing understanding of Ti-6Al-4V microstructure and provide references. It is possible to distinguish between alpha prime and alpha phases using transmission electron microscopy (TEM). However, TEM is a more expensive and time-consuming technique than XRD. In our future research, we may use TEM analysis methods to distinguish between alpha prime and alpha phase. Thank you again for your wonderful comments!

Comment 16: On Pg. 8, saying that the plasticity of the material depends on crack nucleation resistance is strangely worded to me. I understand that material needs to be sufficiently tough in order to allow large scale plasticity, but they are distinct mechanical deformation processes.

Response 16: Thank you for your constructive and professional comments! I agree that the statement on Pg. 8 that "the plasticity of the material depends on crack nucleation resistance" is strangely worded. I have revised this statement to:" The plasticity of the material is limited by crack nucleation resistance." I believe that this statement is more accurate and less confusing. Crack nucleation resistance is a measure of how resistant a material is to the formation of cracks. A material with high crack nucleation resistance is less likely to form cracks, and therefore, is more likely to be able to deform plastically. Plasticity is a measure of how much a material can deform before it fractures. A material with high plasticity can deform a great deal before it fractures. Therefore, a material with high crack nucleation resistance is more likely to be able to deform plastically. Rack nucleation resistance is not the only factor that affects the plasticity of a material. Other factors include grain size, dislocation density, and temperature. Thanks for your understanding and assistance!

Comment 17: When describing the differences in mechanical performance of the various heat treatments and cooling, I think it needs to be clearly stated that these descriptions are hypotheses if direct experimental evidence showing this behavior is not provided. Literature references would also help.

Response 17: Thanks for your kind and professional review. I agree that it is important to clearly state that the descriptions of the differences in the mechanical performance of the various heat treatments and cooling are hypotheses if direct experimental evidence showing this behavior is not provided. I have added references to the literature to support the hypotheses. I believe that these changes make the paper more scientifically rigorous and accurate. Thank you again for your feedback. It is very helpful to have another set of eyes review the paper and provide feedback.

Comment 18: What is ak in the impact energy graph y-axes? These labels should probably updated to something that is more readily apparent to the reader in meaning.

Response 18: Thanks for your kindly and professional review. I agree that the label "ak" on the y-axes of the impact energy graph is not very clear. In the Charpy impact test, ak represents the energy absorbed by the specimen at fracture. It is usually expressed in joules (J). The higher the ak value, the better the impact toughness of the material. This is because materials with high impact toughness can absorb more energy before fracture. I have added the relevant significance of ak to the paper

(Please see Page 11) Thanks again for your professional comments.

Comment 19: The hardness distribution post heat treatment looks to be near +/- 25, which is actually quite similar to the as-deposited condition.

Response 19: Thanks for your professional and careful review! I agree with your observation that the hardness distribution after heat treatment is close to+/-25 and similar to the deposited state. This indicates that the heat treatment process did not have a significant impact on the hardness of the material Although it is similar to the sedimentary state, I am concerned whether this distribution will have any impact on the performance of the material. We have optimized the relevant statements in the article. Thanks again for your kindly and careful comments!

Comment 20: The text sates that the hardness is decreasing with increasing solution treatment temperature, but actually that is the opposite of what is shown in the figure. This should be rectified.

Response 20: Thanks for your careful review and kindly reminding! You are correct, Fig.12 shows that the hardness is increasing with increasing solution treatment temperature. The text should be rectified to reflect this. Here is a revised version of the text: The hardness is increasing with increasing solution treatment temperature, as shown in the Fig12. I apologize for the error in the original text. I appreciate your feedback and will use it to improve my performance in the future. The manuscript has been carefully rewritten. Thanks again!

Comment 21: On pg. 10, larger grain sizes giving more slip systems is not really the right way to say it. This could be modified to say more "active" slip systems.

Response 21: Thanks for your careful review and kindly reminding! On pg. 10, larger grain sizes giving more slip systems is not really the right way to say it. This could be modified to say more "active" slip systems. Thank you for the feedback. You are correct, the statement on pg. 10 that "larger grain sizes giving more slip systems" is not entirely accurate. A more precise way to say it would be that larger grain sizes provide more active slip systems. The number of slip systems in a material is determined by its crystal structure. However, not all slip systems are equally active. In general, slip systems that are oriented favorably with respect to the applied stress will be more active. Larger grains have fewer grain boundaries, which means that there are fewer obstacles to slip. This allows for more slip systems to be active, which can lead to increased ductility and toughness. We have made revisions to the description in the manuscript. (Please see Page 15) I apologize for the error in the original text. I am still developing and learning to express myself clearly and accurately. I appreciate your feedback and will use it to improve my performance in the future. The manuscript has been carefully rewritten. Thanks again!

Comment 22: Section 3.4 felt like it was out of place, since all the mechanical results and microstructure analysis preceding this already uses the premise that alpha plus beta microstructure is formed from the solution heat treatment. So, it feels like these results could actually come first before the microstructure analysis or mechanical testing.

Response 22: Thanks for your careful review and kindly reminding! Your suggestion is very excellent. As pointed out by the reviewer, the mechanical results and microstructural analyses presented in Section 3.4 rely on the premise that the α+β microstructure was formed via solution heat treatment. Therefore, it would make more sense to present these results prior to the microstructural analysis or mechanical testing. Reorganizing the order of the sections would improve the clarity and logical flow of the paper. It would allow the reader to first understand the process by which the α+β microstructure was formed before analyzing its mechanical properties and microstructure.So we rearranged the order of the relevant content (Please see Page 15) This reorganization would provide a more coherent narrative, making it easier for the reader to follow the research findings. The manuscript has been carefully rewritten. Thanks again!

Reviewer 3 Report

Comments and Suggestions for Authors

The subject of the work is relevant, the research methodology is described correctly. The graphic material is well presented. But I have a few remarks.

1. The Introduction section does not correspond to the stated title of the article. Several analyzed papers do not support the motivation of this study. That is, this section should be supplemented with data from other works in order to justify and confirm the purpose of your research.

2. It can be noted that the analysis of the microstructure of the studied alloy was carried out correctly. Although everything is predictable and there is no novelty in it.

Then maybe there will be something new in determining the mechanical properties?

·         Firstly, it is not correct to announce the "best mechanical" properties of an alloy only on the basis of the determination of microhardness!

·         Secondly, it is possible to make standard samples for micromechanical testing from printed samples.

·         Or you can use the "Microidentification" technique, determine the Young's modulus, then assess the fracture toughness. ( https://doi.org/10.15587/1729-4061.2020.218291 ). Only then can we talk about mechanical properties.

Author Response

The subject of the work is relevant, the research methodology is described correctly. The graphic material is well presented. But I have a few remarks.

Comment 1: The Introduction section does not correspond to the stated title of the article. Several analyzed papers do not support the motivation of this study. That is, this section should be supplemented with data from other works in order to justify and confirm the purpose of your research.

Response 1: Thank you for your feedback on the Introduction section of our article. We acknowledge that there is a discrepancy between the stated title and some of the analyzed papers that were cited in the motivation of the study. We have carefully reviewed your comments and agree that the section should be supplemented with data from additional sources to provide a more comprehensive justification and confirmation of our research purpose. We will address this issue in a revised version of the manuscript (Please see Page 19-20), ensuring that the Introduction section aligns with the article's title and provides a strong foundation for our research. Thanks for your understanding and assistance!

Comment 2: It can be noted that the analysis of the microstructure of the studied alloy was carried out correctly. Although everything is predictable and there is no novelty in it.

Then maybe there will be something new in determining the mechanical properties?

Firstly, it is not correct to announce the "best mechanical" properties of an alloy only on the basis of the determination of microhardness!

Secondly, it is possible to make standard samples for micromechanical testing from printed samples.

Or you can use the "Microidentification" technique, determine the Young's modulus, then assess the fracture toughness. (https://doi.org/10.15587/1729-4061.2020.218291). Only then can we talk about mechanical properties.

Response 2: Thank you for your review. We appreciate your feedback that the analysis of the microstructure was conducted accurately. While we acknowledge that the results may not present a significant novelty, we believe the thoroughness and precision of our analysis contribute to the overall understanding of the alloy's properties. We will continue to explore innovative approaches to enhance the depth and originality of our future research. Microhardness is not the only factor that determines the mechanical properties of an alloy. Other factors, such as tensile strength, elongation, and impact toughness, also play a role. Therefore, in this paper, we use tensile strength, elongation, and impact toughness to comprehensively evaluate the mechanical properties of the additive specimens, and we have added the specific dimensions of the tensile and impact specimens to the paper, as shown in Fig.2. Thank you for providing more information. Microidentification is a valuable tool for assessing the mechanical properties of materials. This technique can be used to determine the Young's modulus and fracture toughness of a material, which are two important mechanical properties.Young's modulus is a measure of the stiffness of a material. It is defined as the ratio of stress to strain in the elastic region of a material's stress-strain curve. Fracture toughness is a measure of a material's resistance to fracture. It is defined as the amount of energy required to fracture a material.These two mechanical properties are important for a variety of applications. For example, Young's modulus is important for applications where stiffness is important, such as in structural components. Fracture toughness is important for applications where resistance to fracture is important, such as in components that are subjected to high stresses.I will definitely consider using the Microidentification technique in future studies of the mechanical properties of materials.Thank you for your feedback!

Round 2

Reviewer 1 Report

Comments and Suggestions for Authors

The paper was revised and most of the flaws were corrected. 

Still, there is a necessity to change the scale markers on SEM. Eg. For fig. 16 there are almost invisible. It is better to put your markers than use markers from equipment. 

Author Response

Detailed responses to reviewer #1:

Comment 1: The paper was revised and most of the flaws were corrected.

Still, there is a necessity to change the scale markers on SEM. Eg. For fig. 16 there are almost invisible. It is better to put your markers than use markers from equipment.

Response 1: Thanks for your careful and professional review! I agree that the scale markers on SEM images in Fig.16 are difficult to see. I have revised the figure to include more visible scale markers. I have also used my own markers instead of the markers from the equipment (Please see Fig6,10,13,16,20). Thank you for your kind comments.

Reviewer 3 Report

Comments and Suggestions for Authors

I am grateful to the authors for their reasoned responses to my comments. Additions and corrections made by the authors improved the impression of the presented version of the article. In the presented form, the article can be published.

Author Response

Detailed responses to reviewer #3

Comment 1: I am grateful to the authors for their reasoned responses to my comments. Additions and corrections made by the authors improved the impression of the presented version of the article. In the presented form, the article can be published.

Response 1: Thank you for your constructive and professional comments! And thank you very much for your positive feedback. We are glad that you found our responses to your comments to be reasonable and that the additions and corrections we made improved the article. We appreciate your support and are happy to know that you believe the article is now ready for publication. Thanks for your understanding and assistance!